# Advancing *Toxoplasma gondii* multiplex serology

Rima Jeske,[1] Nico Becker,[1,2] Lea Kroeller,[1,3] Alexander J. Mentzer,[4] Nicole Brenner,[1] Edward Guy,[5] Tim Waterboer,[1] Julia Butt[1]

**ABSTRACT** *Toxoplasma gondii* is a highly prevalent pathogen causing zoonotic infections with significant public health implications. Yet, our understanding of long-term consequences, associated risk factors, and the potential role of co-infections is still limited. Seroepidemiological studies are a valuable approach to address open questions and enhance our insights into *T. gondii* across human populations. Here, we present substantial advancements to our previously developed *T. gondii* multiplex serology assay, which is based on the immunodominant antigens SAG1 and P22. While our previous bead-based assay quantified antibody levels against multiple targets in a high-through-put fashion requiring only a small sample volume, impaired assay characteristics emerged in sample dilutions beyond 1:100 and when being transferred to magnetic beads. Both are now critical for inclusion in large-scale seroprevalence studies. Using the truncated versions, SAG1D1 and P22trunc, significantly enhanced signal-to-noise ratios were achieved with almost perfect concordance with the gold-standard Sabin–Feldman dye test. In sample dilutions of 1:100, the diagnostic accuracy of SAG1D1 and P22trunc reached sensitivities (true positive rates) of 98% and 94% and specificities (true negative rates) of 93% and 95%, respectively. Importantly, performance metrics were reproducible in a 1:1,000 sample dilution, using both magnetic and nonmagnetic beads. Thresholds for seropositivity were derived from finite mixture models and performed equally well as thresholds by receiver operating characteristic analysis. Our improved multiplex serology assay is therefore able to generate robust and reproducible performance metrics under various assay conditions. Inclusion of *T. gondii* antibody measurements with other pathogens, in multiplex serology panels will allow for large-scale seroepidemiological research.

**IMPORTANCE** *Toxoplasma gondii* is a pathogen of significant public health concern due to its widespread prevalence and zoonotic potential. However, our understanding of key aspects, such as risk factors for infection and disease, potential outcomes, and their trends, remains limited. Seroepidemiological studies in large cohorts are invaluable for addressing these questions but remain scarce. Our revised multiplex serology assay equips researchers with a powerful tool capable of delivering *T. gondii* serum antibody measurements with high sensitivity and specificity under diverse assay conditions. This advancement paves the way for the integration of *T. gondii* antibody measurements into multi-pathogen multiplex serology panels, promising valuable insights into public health and pathogen interactions.

**KEYWORDS** *Toxoplasma gondii*, multiplex serology, public health, seroepidemiology, high-throughput serology

T oxoplasma gondii is an obligate intracellular parasite that is capable of infecting a wide range of host species, including humans (1). According to seroepidemiological studies, approximately one-third of the global human population is infected, with particularly high seroprevalences in parts of Africa, Oceania, South America and Europe (2, 3).

Address correspondence to Rima Jeske, R.Jeske@dkfz.de.

Tim Waterboer and Julia Butt contributed equally to this article.

The authors declare no conflict of interest.

Infections are acquired by ingestion of tissue cysts in undercooked meat from infected animals or by ingestion of oocysts from contaminated food or water. The major source of these environmental contaminations is suggested to be domestic cats due to the excretion of oocysts in their feces (4, 5). Acute infection with *T. gondii* is most commonly asymptomatic and may transition into a latent chronic state, which is kept in check by the immune system (6). In congenitally infected newborns and immuno-compromised individuals, for example, tissue-transplanted or HIV-infected individuals, acute or chronic *T. gondii* may escape immune control, resulting in a range of severe or life-threatening conditions, for example, encephalitis (1).

Given its high prevalence and the potentially severe outcomes in some individuals, the interest in studying long-term effects of *T. gondii* infections is high, especially as reports about associations with neurobehavioral changes and disorders, neurological conditions, and different types of cancer accumulate (7–10).

Focusing on seroepidemiological research, we previously developed a bead-based multiplex serology assay incorporating the two immunodominant *T. gondii* antigens, SAG1 (P30) and P22 (SAG2A). It was validated against a reference population tested with the gold-standard Sabin–Feldman dye test and achieved a sensitivity of 92.2% and a specificity of 92.0%. The multiplex serology assay has been used to simultane-ously determine antibody responses against *T. gondii* and other infectious agents in a high-throughput fashion (11–14). Unfortunately, we observed a reduced sensitivity at the 1:1,000 sample dilution now being utilized, which potentially caused weakening of epidemiological associations (12). Since 1:1,000 is the most common dilution in multi-pathogen studies, the exclusion of *T. gondii* from further studies was considered.

Here, we present how we addressed and overcame these limitations by exchanging the *T. gondii* antigens SAG1 and P22 for the truncated versions SAG1D1 and P22trunc, potentially increasing the accessibility of known relevant epitopes (15–17). We deter-mined the diagnostic accuracy in two common sample dilutions (1:100 and 1:1,000) using nonmagnetic and magnetic beads. The latter are of special importance to prepare the automatization of large-scale studies, as magnetic beads can be integrated into liquid handling platforms more easily.

Furthermore, we generated thresholds for seropositivity based on finite mixture models and compared them to thresholds derived from classical receiver operating characteristic (ROC) analyses. This distribution-based method to determine thresholds for seropositivity could be helpful in case antibody responses are systemically affected by experiment-specific or study-specific conditions, for example, bead types, sample dilutions, or population differences (18).

Overall, the improved *T. gondii* multiplex serology assay we present here enables its inclusion into future multi-pathogen disease surveillance studies. These will provide critical sources of evidence to underpin infection risk assessment and thereby inform future public health policy and decision-making.

## MATERIALS AND METHODS

### Reference population

Serum samples for the validation of the *T. gondii* multiplex serology assay were obtained through routine testing and kindly provided by the Toxoplasma Reference Unit of Public Health Wales (Prof. Edward Guy). All samples were tested with the Sabin–Feldman dye test applying a cutoff level of 2 IU/ml. The assay was calibrated against the WHO ToxoG International Standard Preparation (19).

From the original 198 serum samples used for the initial validation of SAG1 and P22, a total of 162 samples (121 samples with a seropositive and 41 samples with a seronegative reference status for *T. gondii*) had sufficient residual volume for the present re-validation (14). All samples had been stored at −20℃.

## T. gondii antigens

All *T. gondii* antigens were expressed as recombinant fusion proteins in *Escherichia coli* BL21, except for P22 which was expressed in *E. coli* BL21 Rosetta. SAG1 and P22 were derived from the initial multiplex serology assay, while SAG1D1, SAG1C, SAG1Pept, and P22trunc were produced for the here presented re-validation (Table 1). Their design is based on structural information deposited in UniProt, as well as published studies on immunogenicity and epitopes (15–17).

Protein sequences were obtained from UniProt, and EMBOSS backtranseq was used for backtranslation and codon optimization (20, 21). The respective DNA sequences were synthesized and subcloned into the pGEX-4T3tag vector by Eurofins genomics. This vector encoded an N-terminal glutathione S-transferase (GST), as well as a C-terminal peptide comprising 11 amino acids of the large T-antigen of the simian virus 40 (22). To verify DNA and protein integrity, plasmids were sequenced, and recombinant proteins were characterized by Western blots and enzyme-linked immunosorbent assays.

Successful full-length expression was accomplished for all of the fusion proteins; however, SAG1C and SAG1Pept did not show promising results in serological pilot experiments. While SAG1Pept turned out to be nonimmunogenic, SAG1C antibody levels were high, but not discriminative for *T. gondii* infection status (Fig. S1). Hence, the multiplex serology results for these two antigens were not further included in the results.

In addition to the *T. gondii* antigens, a series of common control antigens was prepared accordingly, to check for variation across experiments. These were the envelope glycoproteins gE (ORF68) and gI (ORF67) of the varicella zoster virus and the capsid protein VP1 of the human polyomavirus 6 (23, 24).

## Multiplex serology

Multiplex serology was performed as described previously (14, 22). Briefly, recombinant GST-fusion proteins were affinity-purified on color-coded SeroMAP (nonmagnetic) or MagPlex (magnetic) beads (Luminex Corp., Austin, TX, USA), which had been derivatized with glutathione-casein. Serum samples were diluted in preincubation buffer, incubated for 1 h under agitation and added to the bead mix, which contained the respective antigens of interest, in a final serum dilution of 1:100 or 1:1,000.

After another incubation for 1 h, bound serum antibodies were labeled with biotinylated secondary anti-human IgG/IgM/IgA antibodies (goat anti-human IgM/IgA/IgG, Jackson ImmunoResearch, West Grove, PA, USA) and Streptavidin-R-Phycoerythrin (MossBio, Pasadena, MD, USA).

A Luminex 200 Analyzer (Luminex Corp., Austin, TX, USA) was used to determine the bead sort and quantify bound serum antibodies, given as median fluorescence intensity (MFI). For magnetic MagPlex beads, a Luminex FLEXMAP 3D was used, which generated 1.7 times higher MFI values (according to manufacturer) due to hardware/photo multiplier differences.

Assigning different antigens to spectrally distinguishable bead sorts allows quantifying antibody responses against all antigens simultaneously in a single reaction. Yet, the full-length versions (SAG1 and P22) were measured separately from the truncated versions (SAG1D1 and P22trunc) to prevent competing antibody responses due to large

**TABLE 1** Characterization of *T. gondii* antigens

| Antigen | Amino acids | UniProt reference | Alternative names |
| --- | --- | --- | --- |
| SAG1 | 31[a]–349 | A0A125YP09 | P30, SAG-related sequence SRS29B |
| SAG1D1 | 47–178 | | |
| SAG1C | 301–333 | | |
| SAG1Pept | 103–116 | | |
| P22 | 9[a]–187 | A0A125YIJ3 | SAG2A, SAG-related sequence SRS34A |
| P22trunc | 27–173 | | |

[a]Signal peptides were removed.

overlapping regions. Common control antigens were included in both multiplex serology panels to verify comparability between the measurements (Fig. S2).

## Statistical evaluation

Raw MFI values were corrected for bead-specific background values and serum-specific GST-background signals to obtain net MFI values. They were characterized by medians and interquartile ranges (IQRs). Signal-to-noise ratios were obtained by dividing the median net MFI of individuals with a positive reference status (signal) by the median net MFI of those with a negative reference status (noise).

To compare antibody responses between two antigens, linear regression was applied, and numerical correlation was assessed by Pearson's $r$. It was interpreted as follows: $r < 0.30$, slight correlation; $0.3 < r < 0.50$, moderate correlation; $0.5 < r < 0.80$, strong correlation; $0.80 < r$, very strong correlation.

In order to compare multiplex serology results to the gold-standard Sabin–Feldman dye test, continuous MFI values were dichotomized using antigen-specific thresholds. These thresholds were derived from finite mixture models, using an expectation-maximization (EM) algorithm to fit a bimodal distribution after log-transforming MFI values (25, 26). Thresholds corresponded to the local minima of the density curves. Sensitivities and specificities were determined in a next step based on the reference status of the sera.

To evaluate algorithm performance, the thresholds were compared with those derived from the classical maximization of Youden's index (ROC analysis). Paired ROC curves were considered significantly different if the $P$ value was below 0.05 applying DeLong's test to compare area under the curve (AUC) values.

Concordance with the reference assay was assessed using Cohen's kappa ($\kappa$) and interpreted as follows: $\kappa < 0.20$, slight agreement; $0.21 < \kappa < 0.40$, fair agreement; $0.41 < \kappa < 0.60$, moderate agreement; $0.61 < \kappa < .80$, substantial agreement; $0.81 < \kappa$, almost perfect agreement. Additionally, specificities and sensitivities for each threshold were calculated. The respective 95% confidence intervals (95% CIs) were determined using 2,000 stratified bootstrap replicates (27).

All analyses were conducted using R 4.1.3 and the packages pROC and mixsmsn (25, 27). Figures were prepared with GraphPad Prism V. 9 (GraphPad Software, Inc., San Diego, CA, USA).

## RESULTS

### Antibody levels against *T. gondii* antigens vs reference assay status

In a serum dilution of 1:100, individuals with a positive reference status for *T. gondii* reached median antibody levels of 265 MFI (IQR: 165–407 MFI) and 548 MFI (IQR: 238–1,240 MFI) against the full-length antigens SAG1 and P22, respectively. Antibody levels against the truncated antigens SAG1D1 and P22trunc achieved notably higher antibody levels with medians of 991 MFI (IQR: 566–1,648 MFI) and 1,026 MFI (IQR: 454–2,133 MFI), respectively. At the same time, antibody levels against the truncated antigens were decreased in individuals with a negative *T. gondii* reference status with medians of 49 MFI (IQR: 26–96 MFI) and 15 MFI (IQR: 7–27 MFI), respectively, compared to antibody levels against the full-length antigens SAG1 and P22, which reached medians of 67 MFI (IQR: 44–119) and 76 MFI (IQR: 41–123), respectively (Table 2).

Compared to the respective full-length antigen, the signal-to-noise ratio was increased 5.1-fold for SAG1D and 9.5-fold for P22trunc (Table 2; Fig. 1). Yet, numerical correlations between the paired antigens were strong and very strong with Pearson's $r$ values of 0.76 and 0.88, respectively (Fig. S3).

In a serum dilution of 1:1,000, antibody levels against *T. gondii* antigens were generally lower, as expected. While the median antibody levels against SAG1 and P22 were 141 MFI (IQR: 57–250 MFI) and 187 MFI (IQR: 95–513 MFI) in seropositive individuals, they reached 924 MFI (IQR: 296–1,813 MFI) against SAG1D1 and 481 MFI (IQR: 165–1,406 MFI) against P22trunc. In seronegative individuals, the respective antibody levels were

**TABLE 2** Median antibody responses and IQRs against *T. gondii* antigens in individuals with positive (*T. gondii* pos.) and negative (*T. gondii* neg.) reference statuses determined in two serum dilutions[b]

| | | *T. gondii* pos. [MFI (IQR)] | *T. gondii* neg. [MFI (IQR)] | Median signal-to-noise increase [fold-change[a]] | AUC (95% CI) | *P* value |
|---|---|---|---|---|---|---|
| 1:100 | SAG1 | 265 (165–407) | 67 (44–119) | | 0.88 (0.82–0.94) | |
| | SAG1D1 | 991 (566–1,648) | 49 (26–96) | 5.1 | 0.98 (0.94–1.00) | <0.001 |
| | P22 | 548 (238–1,240) | 76 (41–123) | | 0.93 (0.89–0.97) | |
| | P22trunc | 1,026 (454–2,133) | 15 (7–27) | 9.5 | 0.98 (0.97–1.00) | <0.01 |
| 1:1,000 | SAG1 | 141 (57–250) | 18 (10–25) | | 0.93 (0.89–0.98) | |
| | SAG1D1 | 924 (296–1,813) | 13 (5–21) | 9.1 | 0.98 (0.95–1.00) | <0.01 |
| | P22 | 187 (95–513) | 16 (11–31) | | 0.94 (0.91–0.99) | |
| | P22trunc | 481 (165–1,406) | 4 (1–7) | 10.3 | 0.99 (0.99–1.00) | <0.05 |

[a]Signal-to-noise ratio of truncated antigen divided by the signal-to-noise ratio of the full-length antigen.
[b]MFI: median fluorescence intensity; IQR: interquartile range; AUC: area under the curve; CI: confidence interval

18 MFI (IQR: 10–265 MFI), 16 MFI (IQR: 11–31 MFI), 13 MFI (IQR: 5–21 MFI), and 4 MFI (IQR: 1–7 MFI). Hence, the signal-to-noise ratio was increased 9.1-fold for SAG1D1 and 10.3-fold for P22trunc and even more prominent than in the 1:100 serum dilution (Table 2; Fig. 1). Correlations were also high and reached Pearson's *r* values of 0.86 and 0.95, respectively (Table 2).

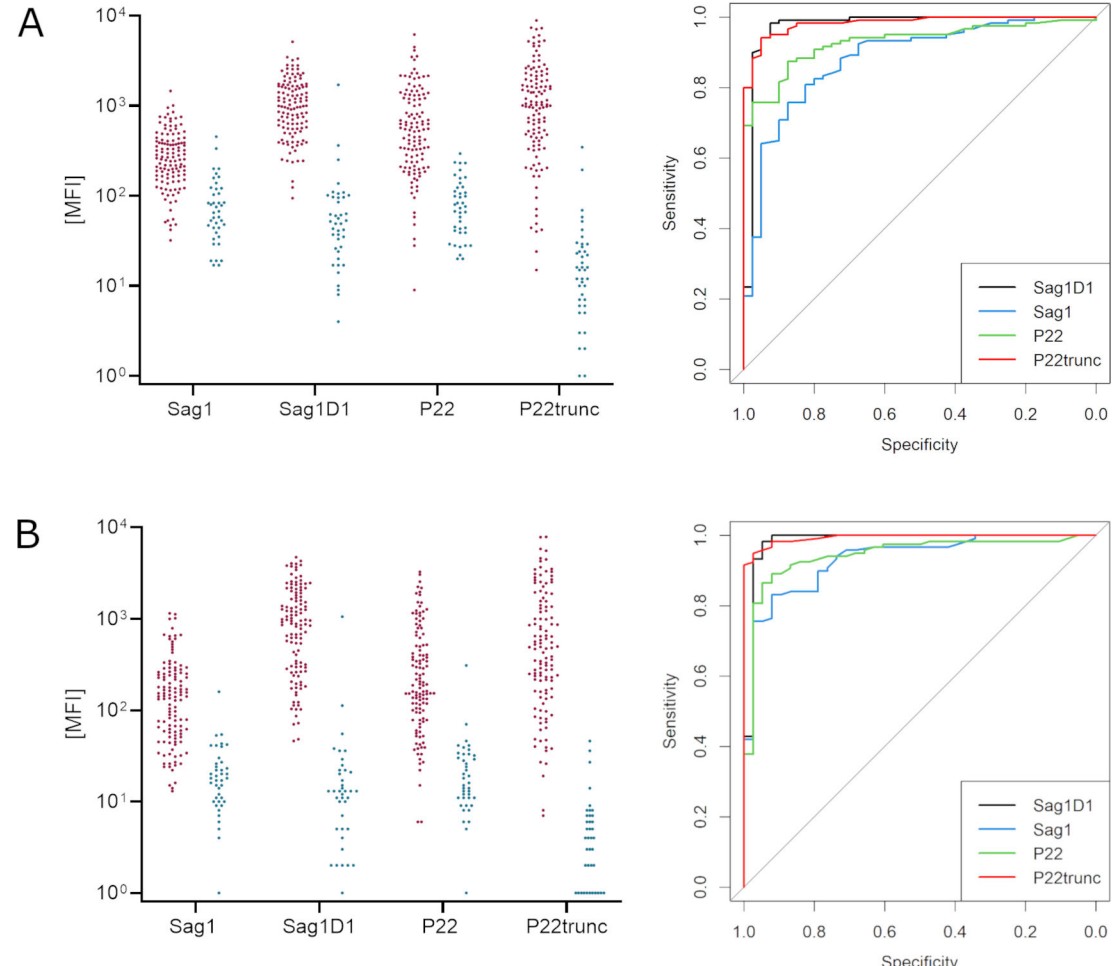

**FIG 1** Antibody levels against *T. gondii* antigens in seropositive (magenta) and seronegative individuals (blue), as well as their eligibility to distinguish them visualized as ROC curves in a serum dilution of 1:100 (A) and 1:1,000 (B).

In order to compare if full-length or truncated antigens can be equally used to distinguish seropositive from seronegative individuals, ROC analyses were performed. While the full-length antigens SAG1 and P22 reached good AUCs between 0.88 and 0.94, the truncated versions significantly outperformed them with AUCs achieving values of 0.98 and 0.99 and nonoverlapping 95%CI (Fig. 1).

## Nonmagnetic vs magnetic beads

The *T. gondii* multiplex serology assay was originally validated using nonmagnetic SeroMAP polystyrene beads. For large seroepidemiological studies, magnetic beads can be of advantage, though, as they facilitate the integration into liquid handling platforms. Hence, we compared antibody levels against the new truncated *T. gondii* antigens quantified with nonmagnetic and magnetic beads in a serum dilution of 1:1,000.

Using nonmagnetic beads, median antibody responses reached 924 MFI (IQR: 296–1,813 MFI) against SAG1D1 and 481 MFI (IQR: 164–1,479 MFI) against P22trunc in seropositive individuals (Table 2). On magnetic beads, antibody levels were increased 6.2 times with a median reaching 5,534 MFI (IQR: 1,850–10,183 MFI) and 4.0 times with a median of 2,230 MFI (IQR: 730–5,714 MFI), respectively, as extracted from the regression line slopes. This led to all antibody responses in seropositive individuals to exceed the lower limit of quantitation of 30 MFI (Fig. 2).

Antibody levels in seronegative individuals were accordingly elevated with median antibody responses of 91 MFI (IQR: 52–164 MFI) against SAG1D1 and 21 MFI (IQR: 7–39 MFI) against P22trunc. Using nonmagnetic beads, these values were 13 MFI (IQR: 5–21 MFI) and 4 MFI (IQR: 1–7 MFI), respectively. Yet, numerical correlations between the two bead types were very strong with Pearson's *r* values of 0.92 (95% CI: 0.89–0.94) and 0.94 (95% CI: 0.91–0.95), respectively.

## Classical vs fitted thresholds and definition of *T. gondii* seropositivity

As demonstrated, numerical antibody levels vary due to different sample dilutions and bead types. Additionally, they can be influenced by population differences, sample handling, and readout analyzers. Thus, prespecifying thresholds for seropositivity can be challenging (18). To investigate if thresholds solely based on the overall distribution are concordant with ROC analysis-based thresholds (a), we fitted finite mixture models assuming two underlying subpopulations, which were either normal distributions (b) or skew-normal distributions (c). They are summarized alongside sensitivities, specificities, and agreement with the reference assay in Table 3.

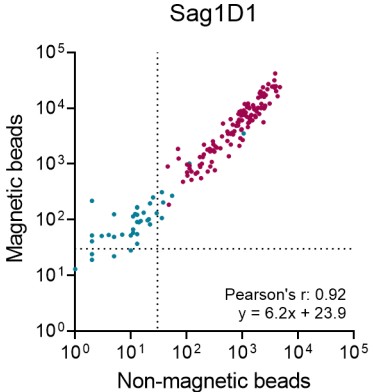
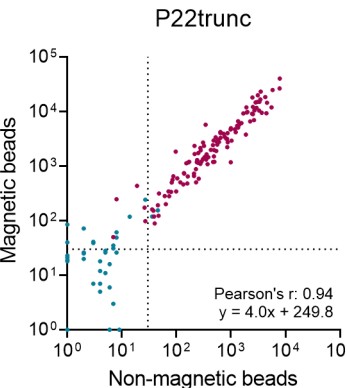

FIG 2 Antibody responses against the *T. gondii* antigens SAG1D1 and P22trunc measured in multiplex serology assays using nonmagnetic beads and magnetic beads. Dotted lines represent the lower limit of quantitation (30 MFI) using a 1:1,000 sample dilution. Magenta-colored dots represent individuals with a positive reference status for *T. gondii*, while blue dots represent individuals with a negative reference status for *T. gondii*.

**TABLE 3** Diagnostic accuracy applying threshold values based on (a) maximizing Youden's index after ROC analysis, (b) the local minimum of two mixed normal distributions, and (c) the local minimum of two mixed skew-normal distributions[b,c]

| Serum dilution | Bead type | P22trunc | | | | SAG1D1 | | | |
| --- | --- | --- | --- | --- | --- | --- | --- | --- | --- |
| | | Threshold (MFI) | Se (95% CI) | Sp (95% CI) | Cohen's κ (95% CI) | Threshold (MFI) | Se (95% CI) | Sp (95% CI) | Cohen's κ (95% CI) |
| **ROC analyses** | | | | | | | | | |
| 1:100 | A | 70 | 94 (94–99) | 95 (80–100) | 0.86 (0.77–0.95) | 141 | 98 (21–100) | 93 (69–100) | 0.92 (0.85–0.99) |
| 1:1,000 | A | 37 | 95 (89–100) | 97 (87–100) | 0.81 (0.71–0.91) | 63 | 98 (38–100) | 95 (84–100) | 0.93 (0.86–1.00) |
| 1:1,000 | B | 158 | 95 (87–99) | 98 (88–100) | 0.89 (0.81–0.97) | 394 | 99 (57–100) | 93 (71–100) | 0.93 (0.87–1.00) |
| **Mixture of two normally distributed subpopulations** | | | | | | | | | |
| 1:100 | A | 113 | 93 (79–99) | 95 (83–100) | 0.83 (0.73–0.93) | 180 | 98 (21–100) | 93 (68–100) | 0.90 (0.83–0.98) |
| 1:1,000 | A | 19[a] | 96 (89–100) | 95 (82–100) | 0.81 (0.71–0.91) | 220 | 82 (38–100) | 97 (92–100) | 0.67 (0.55–0.79) |
| 1:1,000 | B | 140 | 96 (90–100) | 95 (83–100) | 0.89 (0.81–0.97) | 350 | 99 (58–100) | 93 (71–100) | 0.93 (0.87–1.00) |
| **Mixture of two skew-normally distributed subpopulations** | | | | | | | | | |
| 1:100 | A | 95 | 93 (79–99) | 95 (83–100) | 0.83 (0.73–0.93) | 165 | 98 (21–100) | 93 (68–100) | 0.90 (0.83–0.98) |
| 1:1,000 | A | 20[a] | 96 (89–100) | 95 (82–100) | 0.81 (0.71–0.91) | 55 | 98 (84–100) | 92 (88–100) | 0.93 (0.86–1.00) |
| 1:1,000 | B | 165 | 94 (93–99) | 98 (90–100) | 0.89 (0.81–0.97) | 290 | 99 (84–100) | 90 (71–100) | 0.92 (0.84–0.99) |

[a]A minimal threshold of 30 MFI was applied corresponding to the lower limit of quantitation.
[b]A, nonmagnetic beads; B, magnetic beads.
[c]Se, sensitivity (%); Sp, specificity (%).

Overall, P22trunc and SAG1D1 show a very good discrimination of seropositive from seronegative individuals in both dilutions, both bead types and any of the thresholds for seropositivity. Concordance with the reference assay is almost perfect with κ values from 0.81 to 0.89 for P22trunc and from 0.90 to 0.93 for SAG1D1. The only exception was the modeled threshold for SAG1D1 in a 1:1,000 dilution presuming two underlying normal distributions (b). Compared to a and c, the threshold of 220 MFI was higher, which led to false-negative classifications and a reduced sensitivity of 82%. Apart from this, sensitivities and specificities were constantly high and reached values up to 98%.

To test if a combination of SAG1N and P22trunc could further improve the overall performance of *T. gondii* multiplex serology, the respective antibody responses were plotted against each other (Fig. 3). With any of the given thresholds for seropositivity, the agreement between the two antigens is very high, and only few samples show a discordant result. Combining the *T. gondii* antigens, an overall seropositivity could be defined as either being seropositive for both antigens or at least one antigen.

Applying ROC analyses-derived thresholds, seropositivity to both antigens sets the specificity to 95% in a 1:100 sample dilution and 100% in a 1:1,000 sample dilution using

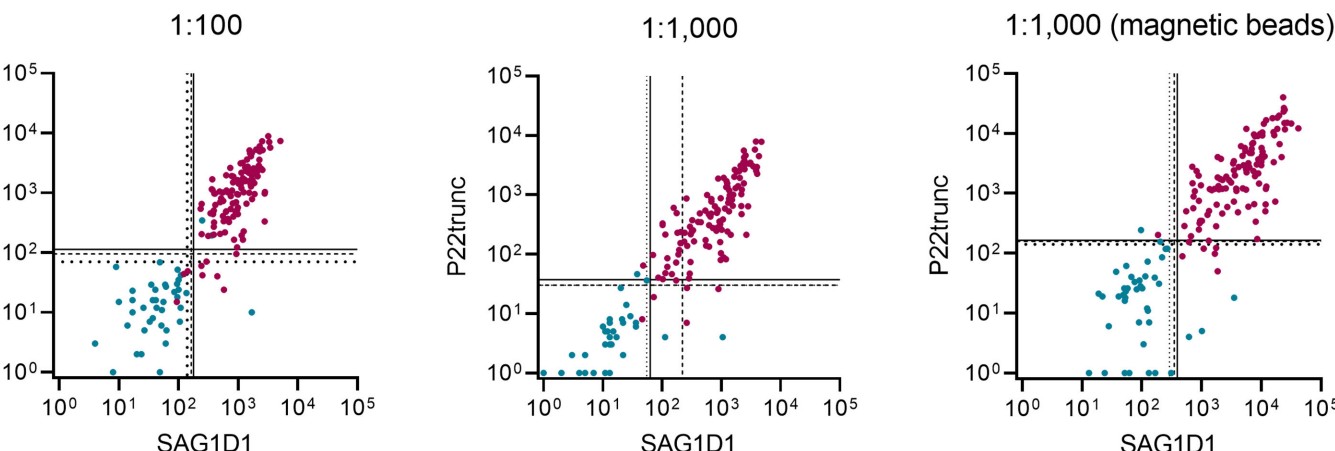

**FIG 3** Antibody levels against the *T. gondii* antigens SAG1D1 and P22trunc in individuals with a seropositive (magenta) and seronegative (blue) reference status. Lines represent proposed thresholds for seropositivity, derived from ROC analysis (solid), fitting a mixture of two normal distributions (dashed) or fitting a mixture of two skew-normal distributions (dotted).

nonmagnetic or magnetic beads. Simultaneously, sensitivities reach 94%, 94%, and 97%, respectively.

If an overall seropositivity for *T. gondii* is defined as being seropositive to at least one of the two antigens, sensitivities increase to 98%, 99%, and 99%, respectively, while specificities decrease to 93%, 93%, and 90%, respectively.

Specificity and sensitivity values for modeled thresholds achieve comparable performances.

## DISCUSSION

We here present recent advancements in our *T. gondii* multiplex serology. By exchanging SAG1 and P22 for their truncated versions SAG1D1 and P22trunc, we substantially increased the magnitude of measured antibody responses by up to 10.3 times. This broadens the range of sample dilutions in which antibodies against *T. gondii* can be reliably determined without losses in sensitivity or specificity. These results were reproducible using different bead types and methods to determine thresholds for seropositivity with *κ* being consistently above 0.9.

Our multiplex serology assay is based on the two immunodominant proteins P22 (SAG2A) and SAG1 (P30), which are expressed in virtually all *T. gondii* strains (28). Both are glycosylphosphatidylinositol-anchored proteins and predominantly found on tachyzoites, the rapidly proliferating life stage of *T. gondii* (29). They are involved in attachment and invasion and, hence, an early target for the immune system. Their potential as serological targets have been recognized early (30, 31). Since then, they have been characterized in more detail.

Macêdo et al. (16) investigated the three-dimensional structure of P22 and described an intrinsically unstructured loop at the C-terminal end of the single-domain protein. *In vitro* experiments revealed that this loop actively suppresses pro-inflammatory responses in cells of the immune system. Generally, unstructured regions are more flexible and often have multiple interaction partners. This attribute could, however, cause problems in serological assays, for example, by sterically shielding off epitopes or aggregating assay components. Hence, we decided to remove the C-terminus but keep a known epitope intact which comprises amino acids 137–141 (17). This truncation significantly increased the amount of bound serum antibodies by 9.5 or 10.3 times in a serum dilution of 1:100 or 1:1,000, respectively. Whether this presumed epitope shielding only occurs in the artificial environment of serological assays or is an actual physiological defense mechanism requires further studies.

The second antigen SAG1 comprises a N-terminal domain D1, a C-terminal domain D2, and an unstructured loop at the C-terminal end (32). Even before these structural domains were described, the N-terminal region was found to harbor most immunodominant epitopes (33). Furthermore, the D1 domain has fewer polymorphisms than the rest of the protein and other SAG proteins, which makes it a stable serological target (32). Using Sag1D1 as an antigen in multiplex serology increased the bound serum antibodies 5.1 and 9.1 times, in a dilution of 1:100 and 1:1,000, respectively. Again, we hypothesize that this was caused by substantially better accessibility of relevant epitopes and/or removing of nonspecific epitopes.

Additionally, we tested the C-terminal end as a potential antigen, as Wang et al. (15) described a highly specific epitope located at amino acids 313–332. In our reference population, however, SAG1C showed no discrimination between sera from individuals with or without a positive reference status for *T. gondii. A*lthough the sequence does not show homologies with any other pathogenic species, this might still be caused by cross-reactive antibodies or by high background values due to the nature of the unstructured loop.

Another peptide comprising amino acids 103–116 (SAG1pept), which comprised an exposed loop between two beta sheets, showed barely any antigenicity in our multiplex serology assay (15). Although the epitope selections were based on the publication by Wang et al., the exact constructs differed slightly, as we tried to account for the predicted

3D structure found in the respective UniProt entries (20). Furthermore, these epitopes were identified using pig serum and might therefore not directly be transferable to humans (15).

An advantage of multiplex serology is that multiple antigens from various pathogens can be combined into a single reaction, which is cost-effective and saves patient material. For a few pathogens, however, this may cause a diminished sensitivity or specificity if the final assay dilution is not ideal. This tradeoff was made for *T. gondii* in a large seroepidemiological study comprising approximately 10,000 participants from the UK Biobank (UKB) (12). Despite a reduced sensitivity in a 1:1,000 sample dilution, association analyses were still robust and consistent with the literature, but effect sizes might have been weakened. Nevertheless, excluding *T. gondii* from similar multi-pathogen studies, which are currently planned, was discussed. Using the truncated *T. gondii* antigens SAG1D1 and P22trunc obviates this consideration, as ROC analyses show that AUCs are very high in both common assay dilutions, 1:100 and 1:1,000.

Furthermore, both antigens perform equally well on magnetic beads with numerical correlation values of 0.92 and 0.94 for Sag1D1 and P22trunc. The principal difference was a systematic increase of the quantitative antibody response using magnetic beads by the factor 6.2 or 4.0, respectively. Increased MFI values have been reported for magnetic beads and were additionally magnified here by using a Luminex FlexMap3D as opposed to a Luminex 200 as output device (12, 34, 35). Multiplex serology is usually robust against these systematic variations for most pathogens. As presented in the UKB assay validation, comparing nonmagnetic beads to magnetic beads resulted in a median ICC (intraclass correlation coefficient) of 0.94 (12). With an ICC of 0.48, *T. gondii* multiplex serology was a strong outlier. As antibody responses are in a low to mid-range, the threshold for seropositivity is prone to systematic assay variation, for example, induced by different bead types. Unfortunately, it is not always possible to control or foresee these variations, as they might be caused by sample preparation, storing conditions, population differences, etc. (18).

Considering that the threshold may need adjustments in order to account for sample-specific characteristics, for example, storage conditions, assay components, nonmagnetic vs magnetic beads, or sample dilutions, a reference-independent approach would be of help to generate reproducible results. Hence, thresholds tailored to the study or experiment are strongly desirable to make seroepidemiological *T. gondii* research more robust. For this purpose, we examined thresholds from fitting finite mixture models to the overall data distribution and compared them to thresholds classically derived from ROC analyses.

Assuming that the bimodal distribution comprises a mixture of two underlying subpopulations, we applied an EM algorithm to fit a mixture of two normal distributions or two skew-normal distributions (26, 36). Sensitivity and specificity values that were derived from applying these thresholds for seropositivity were very similar to those based on classically derived thresholds from ROC analyses. The only exception was modeling the antibody response against SAG1D1 in a 1:1,000 dilution assuming two normal distributions. Due to the high threshold, the sensitivity was decreased to 82%, while it reached 93%–99% for all other modeled thresholds. The specificity was consistently high with values between 90% and 98% for both antigens.

While normal distributions are a classical choice for finite mixture models, skew-normal distributions allow some asymmetry across subpopulations, for example, due to seroconversion or seroreversion. Hence, skew-normal distributions have been proposed as a more suitable option for biological data (26, 36).

In the case of antibodies against *T. gondii*, both models perform well as no major seroconversion or seroreversion is expected. While seroconversion usually takes place within a few days to weeks after infection, seroreversion typically occurs over a longer time period after an infection is eliminated. As *T. gondii* usually persists in a latent form and continuously goes through spontaneous reactivations, antibody levels decline but do usually not disappear (37).

Our results showed that finite mixture models seem promising to generate thresholds for *T. gondii* multiplex serology, which are solely based on the data distribution of the results. Nevertheless, a bimodal data distribution needs to be verified in further upcoming seroepidemiological studies. Modeling thresholds might also be transferrable to other antigens and pathogens if both subpopulations (seropositive and seronegative individuals) are large enough. Similarly, validation studies for all antigens are essential, as antibody kinetics might differ and be more complex than for *T. gondii* due to variables such as reinfection rates, cross-reactivity, population differences, age-dependence, and others (18, 26, 38).

Overall, we achieved substantial advancements in *T. gondii* multiplex serology. By truncating recombinant *T. gondii* antigens, we presumably enhanced the accessibility of relevant epitopes, which led to significantly increased antibody levels and an improved assay performance compared to a reference gold-standard assay. Our advancements will allow integrating *T. gondii* antibody measurements into large multi-pathogen panels, as demonstrated for other pathogens in the UKB and the China Kadoorie Biobank (12, 39). This will facilitate seroepidemiological research which is crucial for disease surveillance, risk assessment, and informing public health interventions.

## AUTHOR AFFILIATIONS

[1]Division of Infections and Cancer Epidemiology, German Cancer Research Center (DKFZ), Heidelberg, Germany
[2]Faculty of Biotechnology, Mannheim University of Applied Sciences, Mannheim, Germany
[3]Faculty of Biosciences, Heidelberg University, Heidelberg, Germany
[4]Centre for Human Genetics, University of Oxford, Oxford, United Kingdom
[5]Toxoplasma Reference Unit, Public Health Wales Microbiology, Swansea, United Kingdom

## AUTHOR ORCIDs

Rima Jeske http://orcid.org/0000-0003-4596-3778

## AUTHOR CONTRIBUTIONS

Rima Jeske, Data curation, Formal analysis, Investigation, Project administration, Writing – original draft | Nico Becker, Investigation, Visualization, Writing – review and editing | Lea Kroeller, Investigation, Resources, Writing – review and editing | Alexander J. Mentzer, Conceptualization, Methodology, Resources, Writing – review and editing | Nicole Brenner, Conceptualization, Writing – review and editing | Edward Guy, Methodology, Resources, Writing – review and editing | Tim Waterboer, Conceptualization, Methodology, Resources, Writing – review and editing | Julia Butt, Conceptualization, Supervision, Validation, Writing – review and editing

## ADDITIONAL FILES

The following material is available online.

### Supplemental Material

**Supplemental figures (Spectrum03618-23-S0001.docx).** Fig. S1 to S3.

### Open Peer Review

**PEER REVIEW HISTORY (review-history.pdf).** An accounting of the reviewer comments and feedback.

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
