## [Reviewer comments · Microbiology Spectrum]

Microbiology Spectrum

Advancing *Toxoplasma gondii* multiplex serology

Rima Jeske, Nico Becker, Lea Kroeller, Alexander Mentzer, Nicole Brenner, Edward Guy, Tim Waterboer, and Julia Butt

Corresponding Author(s): Rima Jeske, Deutsches Krebsforschungszentrum

Review Timeline:

Submission Date:	October 12, 2023
Editorial Decision:	November 22, 2023
Revision Received:	January 17, 2024
Accepted:	January 29, 2024

Editor: Mark Pandori

Reviewer(s): Disclosure of reviewer identity is with reference to reviewer comments included in decision letter(s). The following individuals involved in review of your submission have agreed to reveal their identity: Will Pandori (Reviewer #1); Huda Sahib Abdul-Mohammed Al-Rawazq (Reviewer #2)

Transaction Report:

DOI: <https://doi.org/10.1128/spectrum.03618-23>

Re: Spectrum03618-23 (Advancing *Toxoplasma gondii* multiplex serology)

Dear Dr. Rima Jeske:

Thank you for the privilege of reviewing your work. Below you will find my comments, instructions from the Spectrum editorial office, and the reviewer comments.

Revision Guidelines

Sincerely,
Mark Pandori
Editor
Microbiology Spectrum

Reviewer #1 (Comments for the Author):

I am including a document with all of my specific comments.

Most comments revolve around removing statements about reproducibility and robustness of results, as the authors' own discussion suggests that more testing is needed before conclusions about robustness and reproducibility can be made.

I support all conclusions on the improved assay development.

The other major comment is on clarification on the statistics and modeling done for calculating threshold values. It is unclear if either method used a training set of samples and then tested the model on the remaining set of samples, or if all samples were used to generate the threshold models. I am supportive of reproducibility and robustness claims if the analysis is actually done in this way and was just not clear to me.

Figures are well prepared.

Please note a few comments on background and some grammar suggestions.

Reviewer #2 (Comments for the Author):

Dear Author Very good paper but need to some changes,

* Add Key words after Abstract.

* In Title Gondii antigens page 5 line 123 from (These envelope glycoproteins to line 125 delete .

* Add conclusion after Discussion and in Abstract.

* Add Author affiliation.

* Add Author Orchid.

* Add Funding.

* Add Author Contributions.

Thank yo

Goal: Sero-epidemiological studies are a valuable approach to address open questions about risk factors, potential outcomes, trends, and co-infections in large cohorts of people.

Method: *T. gondii* multiplex serology approach, which is based on the immunodominant antigens SAG1 and P22.

Conclusions:

1. Using the truncated versions, SAG1D1 and P22trunc, significantly enhanced signal-to-noise ratios were achieved, and with almost perfect concordance with the gold-standard Sabin Feldman dye test. **(Yes)**
2. In sample dilutions of 1:100, the diagnostic accuracy of SAG1D1 and 29 P22trunc reached sensitivities of 98% and 94% and specificities of 93% and 95%, respectively. **(Yes)**
3. Importantly, however, these results were reproducible in a 1:1,000 sample dilution, using both magnetic and non-magnetic beads, and across multiple methods to determine thresholds for seropositivity, including receiver operating characteristic (ROC) analysis and finite mixture models. **(Yes)**
4. Our improved multiplex serology assay is therefore able to generate robust and **reproducible results** under various assay conditions, thus **enabling inclusion of *T. gondii* antibody measurements with other pathogens**, in large multiplex serology panels for sero-epidemiological research. **(No)**
5. Our revised multiplex serology assay equips researchers with a powerful tool capable of delivering **reliable and reproducible results** under diverse assay conditions. This advancement paves the way for the integration of *T. gondii* antibody measurements into multi-pathogen multiplex serology panels, promising valuable insights into public health and pathogen interactions. **(No)**
6. **Our advancements allow measuring *T. gondii* antibodies reliably in large multi-pathogen panels** and will help to conduct sero-epidemiological research which is crucial for disease surveillance, risk assessment, and informing public health interventions. **(No)**

All sections highlighted red are not supported by the results. The authors state in the discussion section that the results “seem promising” but “require further validation”. Claims about reproducibility and reliability of results, as well as the ability to include this test in larger studies, should not yet be made until those tests or experiments are run and shown to be reliable and reproducible.

Comments:

Line 14: Yet, our understanding of risk factors for infection and disease, potential outcomes and their trends, and the potential role of co-infections is still limited.

- Understanding is not limited. Much is understood from previous studies. Focus more on how increased understanding is still valuable and how it could be used.

Line 29: P22trunc reached sensitivities of 98% and 94% and specificities of 93% and 95%, respectively.

- “Sensitivities” and “specificities” have not previously been defined and it would be helpful to clearly state what they signify or mean.

Line 30: Importantly, however, these results were reproducible in a 1:1,000 sample dilution, using both

- Delete “however”
- Specify which results were reproducible.

Line 55: The major source of these environmental contaminations is suggested to be domestic cats (4, 5).

- Mentioning that infected cats are the definitive host for the parasite and cause this contamination through shedding of oocysts in feces gives the reader a better understanding of how the parasite spreads.

Line 58: In congenitally infected newborns and immunocompromised individuals, e.g. tissue-transplanted or HIV-infected individuals, *T. gondii* may proliferate, resulting in a range of severe or life-threatening conditions, e.g. encephalitis (1).

- Consider including the understanding that both acute and latent chronic infections can escape immune control and be life-threatening to immunocompromised individuals. This would help the transition to, and giving more weight to, the following paragraph.

Line 102: All samples had been stored at -20°C.

- Wonderful. This adds value to the study showing the robustness of the method, allowing multiple freeze/thaw cycles of sample according to previous publications.

Line 96: Serum samples for the validation of the *T. gondii* multiplex serology assay were kindly provided by the Toxoplasma Reference Unit of Public Health Wales (Prof. Edward Guy)

- More information on these samples would be important. For example, where were the samples collected and is it known what strains or “(types I, II, III)” the infected individuals were infected with? I see that some of this information is included in other papers, but referencing a few of the features of this data set is relevant.
- This would also lead into the needed important background on the expression of each of these antigens being tested in the different *T. gondii* strains, as prevalence of the different strains varies greatly among locations and populations being tested. I would strongly suggest citing some of John Boothroyd’s lab’s recent work here for the introduction.

Line 187: Overall, the signal-to-noise ratio was increased for both truncated antigens with MFI values reaching on average 2.9-times higher MFI values for SAG1D1 and 1.6-times for P22trunc compared to the respective full-length versions (Table 2, Figure 1).

- This statement is accurate, but when working with MFI referencing fold changes instead of raw numerical MFI changes can be misleading. For example, the difference in an MFI of 10 and 30 results in a 3-fold difference but only a 20 unit

difference. The difference in an MFI of 50 and 100 results in a 2-fold difference but 50-unit difference. This exact scenario occurs in the next comment. The larger raw unit difference can be what matters most for being able to differentiate between real signal and noise in most experiments. If this is not the case for the models being run, then an explanation of this should be included.

Line 199: Linear regression revealed that the difference in antibody levels between full-length and truncated antigens was even more prominent than in the 1:100 serum dilution, with MFI values being on average 4.4-times increased for SAG1D1 and 2.3-times for P22trunc compared to the respective full-length versions (Table 2, Figure 1). Correlations were also higher and reached Pearson's r values of 0.86 and 0.95, respectively (Table 2).

- It is okay to point out that the fold difference is greater in the 1:1000 dilution, but the raw difference in MFI is only greater for SAG1D1 and not P22trunc. Neglecting this point can be misleading.

Line 217: Using non-magnetic beads, median antibody responses reached 924 MFI (IQR: 296 – 1,813 MFI) against SAG1D1 and 481 MFI (IQR: 164 – 1,479 MFI) against P22trunc in seropositive individuals (Table 2). On magnetic beads, antibody levels were increased 6.2-times with a median reaching 5,534 MFI (IQR: 1,850 – 10,183 MFI) and 4.0-times with a median of 2,230 MFI (IQR: 730 – 5,714 MFI), respectively, as extracted from the regression line slopes. This led to all antibody responses in seropositive individuals to exceed the lower limit of quantitation of 30 MFI (Figure 2).

- The same statement stands for fold vs. raw unit increases.

Discussion

Line 267: These results were reproducible using different bead types and methods to determine thresholds for seropositivity with Cohen's kappa being consistently above 0.9.

- It is clear that ROC analysis was done and finite mixture models were created, but previous papers do go into more detail about these methods than this paper does, and so some questions still remain. Was the model to determine threshold for seropositivity developed using only a subset of the samples and then tested on the remaining samples, or does the model determining the threshold for seropositivity still need to be tested using new samples to measure how accurately it can determine positive from negative samples? Further statements in the discussion suggest that the model has not been tested on other samples, but if that is incorrect then further explanation is needed.

Line 259: As *T. gondii* usually persists in a latent form, antibody levels decline but do usually not disappear.

- If desired, the authors can support this statement with background on how infection will continually go through small spontaneously reactivations from cysts, making sero-reversion more than exceedingly rare.

Line 362: Although, our results showed that thresholds derived from finite mixture models seem promising for *T. gondii* multiplex serology, the exploratory approach requires further validation in large-scale studies. Modeling thresholds might also be transferrable to other antigens if both subpopulations (seropositive and seronegative individuals) are large enough. Similarly, validation studies for all antigens are essential, as antibody kinetics might differ and be more complex due to reinfection rates, cross-reactivity, population differences, age-dependence, and others (18, 35, 36).

- This statement is accurate, but it contradicts the previously and subsequent conclusion that the method generates “robust and reliable results” that could help conduct sero-epidemiological research. Either training the model on a subset of the already collected data and testing it on the other subset, or ideally testing the current threshold models on new samples, is needed to claim that the method can reliably and robustly generate data usable to conduct sero-epidemiological research. It is also acceptable to remove these claims, or change them and instead focus on the proven improvements on assay sensitivity. This could still include a proposed threshold for sero-positivity that could be tested in future experiments. That is already a significant finding.
- Either delete “and others”, or include “variables such as” before “reinfection rates”.

Advancing *Toxoplasma gondii* multiplex serology

Rima Jeske¹, Nico Becker^{1,2}, Lea Kröller^{1,3}, Alexander J Mentzer^{4,5}, Nicole Brenner¹, Edward
Guy⁶, Tim Waterboer^{1,*}, Julia Butt^{1,*}

¹ Division of Infections and Cancer Epidemiology, German Cancer Research Center (DKFZ),
Heidelberg, Germany

² Faculty of Biotechnology, Mannheim University of Applied Sciences

³ Faculty of Biosciences, Heidelberg University, Heidelberg, Germany

⁴ The Wellcome Centre for Human Genetics, University of Oxford, Oxford, UK

⁵ Big Data Institute, Li Ka Shing Centre for Health Information and Discovery, University of
Oxford, Oxford, UK

⁶ Toxoplasma Reference Unit, Public Health Wales Microbiology, Swansea, United Kingdom

* shared last authorship

Abstract

*Toxoplasma gondii* (*T. gondii*) is a highly prevalent pathogen causing zoonotic infections with
significant public health implications. Yet, our understanding of risk factors for infection and
disease, potential outcomes and their trends, and the potential role of co-infections is still
limited. Sero-epidemiological studies are a valuable approach to address these open questions
in large cohorts of people.

Here, we present substantial advancements to our previously developed *T. gondii* multiplex
serology approach, which is based on the immunodominant antigens SAG1 and P22. While
our previous bead-based assay quantified antibody levels against multiple targets in a high-
throughput fashion requiring only a small sample volume, impaired assay characteristics
emerged in sample dilutions beyond 1:100 and when being transferred to magnetic beads.
Both are now critical for inclusion in large-scale seroprevalence studies.

Using the truncated versions, SAG1D1 and P22trunc, significantly enhanced signal-to-noise
ratios were achieved, and with almost perfect concordance with the gold-standard Sabin-
Feldman dye test. In sample dilutions of 1:100, the diagnostic accuracy of SAG1D1 and
P22trunc reached sensitivities of 98% and 94% and specificities of 93% and 95%, respectively.
Importantly, however, these results were reproducible in a 1:1,000 sample dilution, using both
magnetic and non-magnetic beads, and across multiple methods to determine thresholds for
seropositivity, including receiver operating characteristic (ROC) analysis and finite mixture
models.

Our improved multiplex serology assay is therefore able to generate robust and reproducible
results under various assay conditions, thus enabling inclusion of *T. gondii* antibody
measurements with other pathogens, in large multiplex serology panels for sero-
epidemiological research.

Importance

*T. gondii* is a pathogen of significant public health concern due to its widespread prevalence
and zoonotic potential. However, our understanding of key aspects, such as risk factors for
infection and disease, potential outcomes and their trends, remains limited. Sero-
epidemiological studies in large cohorts are invaluable for addressing these questions, but
remain scarce.

Our revised multiplex serology assay equips researchers with a powerful tool capable of
delivering reliable and reproducible results under diverse assay conditions. This
advancement paves the way for the integration of *T. gondii* antibody measurements into
multi-pathogen multiplex serology panels, promising valuable insights into public health and
pathogen interactions.

Introduction

*Toxoplasma gondii* (*T. gondii*) is an obligate intracellular parasite capable of infecting a wide
range of host species, including humans (1). According to sero-epidemiological studies,
approximately one third of the global human population is infected, with particularly high
seroprevalences in parts of Africa, Oceania, South America and Europe (2, 3).

Infections are acquired by ingestion of tissue cysts in undercooked meat from infected animals
or by ingestion of oocysts from contaminated food or water. The major source of these
environmental contaminations is suggested to be domestic cats (4, 5). Acute infection with
*T. gondii* is most commonly asymptomatic and transitions into a latent chronic state, which is
kept in check by the immune system (6). In congenitally infected newborns and
immunocompromised individuals, e.g. tissue-transplanted or HIV-infected individuals,
*T. gondii* may proliferate, resulting in a range of severe or life-threatening conditions, e.g.
encephalitis (1).

[revised manuscript text omitted]

To compare antibody responses between two antigens, linear regression was applied and
numerical correlation was assessed by Pearson's r . It was interpreted as follows: $r < 0.30$: slight
correlation, $0.3 < r < 0.50$: moderate correlation: $0.5 < r < 0.80$: strong correlation, $0.80 < r$: very
strong correlation.

In order to compare multiplex serology results to the gold-standard Sabin-Feldman dye test,
receiver operating characteristic (ROC) analyses were performed and the area under the curve
(AUC) was reported to estimate the test accuracy. Paired ROC curves were considered
significantly different if the p -value was below 0.05 applying DeLong's test.

In order to calculate concordance with the reference status, continuous MFI values were
dichotomized using antigen-specific thresholds. These thresholds either derived from
maximizing Youden's index (ROC analysis) or finite mixture modeling, using an expectation-
maximization (EM) algorithm to fit a bimodal distribution after log-transforming MFI values (25).
Thresholds corresponded to the local minima of the density curves.

Concordance with the reference assay was assessed using Cohen's kappa (κ) and interpreted
as follows: $\kappa < 0.20$: slight agreement, $0.21 < \kappa < 0.40$: fair agreement, $0.41 < \kappa < 0.60$: moderate
agreement, $0.61 < \kappa < 0.80$ substantial agreement, $0.81 < \kappa$: almost perfect agreement.
Additionally, specificities and sensitivities for each threshold were calculated. The respective
95% confidence intervals (95% CI) were determined using 2000 stratified bootstrap replicates
(26).

All analyses were conducted using R 4.1.3 and the packages pROC and mixsmsn (25, 26).
Figures were prepared with GraphPad Prism 9 (GraphPad Software, Inc., San Diego, CA,
USA).

Results

Antibody levels against *T. gondii* antigens vs. reference assay status

In a serum dilution of 1:100, individuals with a positive reference status for *T. gondii* reached
median antibody levels of 265 MFI (IQR: 165 – 407 MFI) and 548 MFI (IQR: 238 – 1,240 MFI)
against the full-length antigens SAG1 and P22, respectively. Antibody levels against the
truncated antigens SAG1D1 and P22trunc achieved notably higher antibody levels with
medians of 991 MFI (IQR: 566 – 1,648 MFI) and 1,026 MFI (IQR: 454 – 2,133 MFI),
respectively. At the same time, antibody levels against the truncated antigens were decreased
in individuals with a negative *T. gondii* reference status with medians of 49 MFI (IQR: 26 – 96
MFI) and 15 MFI (IQR: 7 – 27 MFI), respectively, compared to antibody levels against the full-
length antigens SAG1 and P22, which reached medians of 67 MFI (IQR: 44 – 119) and 76 MFI
(IQR: 41 – 123), respectively (Table 2).

Overall, the signal-to-noise ratio was increased for both truncated antigens with MFI values
reaching on average 2.9-times higher MFI values for SAG1D1 and 1.6-times for P22trunc
compared to the respective full-length versions (Table 2, Figure 1). Yet, numerical correlations
between the paired antigens were strong and very strong with Pearson's *r* values of 0.76 and
0.88, respectively (Supplementary Figure 3).

In a serum dilution of 1:1,000, antibody levels against *T. gondii* antigens were generally lower,
as expected. While the median antibody levels against SAG1 and P22 were 141 MFI (IQR: 57
195 – 250 MFI) and 187 MFI (IQR: 95 – 513 MFI) in seropositive individuals, they reached 924 MFI
(IQR: 296 – 1,813 MFI) against SAG1D1 and 481 MFI (IQR: 165 – 1,406 MFI) against
P22trunc. In seronegative individuals, the respective antibody levels were 18 MFI (IQR: 10 –
265 MFI), 16 MFI (IQR: 11 – 31 MFI), 13 MFI (IQR: 5 - 21 MFI) and 4 MFI (IQR: 1 - 7 MFI).

Linear regression revealed that the difference in antibody levels between full-length and
truncated antigens was even more prominent than in the 1:100 serum dilution, with MFI values
being on average 4.4-times increased for SAG1D1 and 2.3-times for P22trunc compared to
the respective full-length versions (Table 2, Figure 1). Correlations were also higher
[revised manuscript text omitted]

or sample dilutions, an automated approach would be of help to generate reproducible and
reliable results. Hence, thresholds tailored to the study or experiment are strongly desirable to
make sero-epidemiological *T. gondii* research more robust. For this purpose, we examined
generating thresholds by fitting finite mixture models to the overall data-distribution and
compared them to thresholds classically derived from ROC analyses.

Assuming that the bimodal distribution comprises a mixture of two underlying subpopulations,
we applied an EM algorithm to model a mixture of two normal distributions or two skew-normal
distributions (34, 35). Sensitivity and specificity values that derive from applying these
thresholds for seropositivity were very similar to those based on 'ideal' thresholds derived from
ROC analyses. The only exception was modeling the antibody response against SAG1D1 in
a 1:1,000 dilution assuming two normal distributions. Due to the high threshold, the sensitivity
was decreased to 82%, while it reached 93 - 99% for all other modeled thresholds. The
specificity was consistently high with values between 90% and 98% for both antigens.

While normal distributions are a classical choice for finite mixture models, skew-normal
distributions allow some asymmetry across subpopulations, e.g. due to seroconversion
or -reversion. Hence, skew-normal distributions have been proposed as a more suitable option
for biological data (34, 35).

In the case of antibodies against *T. gondii*, both models perform well as no major
seroconversion or seroreversion is expected. While seroconversion usually takes place within
a few days to weeks after infection, seroreversion typically occurs over a longer time period
after an infection is eliminated. As *T. gondii* usually persists in a latent form, antibody levels
decline but do usually not disappear.

Although, our results showed that thresholds derived from finite mixture models seem
promising for *T. gondii* multiplex serology, the exploratory approach requires further validation
in large-scale studies. Modeling thresholds might also be transferrable to other antigens if both
subpopulations (seropositive and seronegative individuals) are large enough. Similarly,
validation studies for all antigens are essential, as antibody kinetics might differ and be more
complex due to reinfection rates, cross-reactivity, population differences, age-dependence,
and others (18, 35, 36).

Overall, we achieved substantial advancements in *T. gondii* multiplex serology. By truncating
recombinant *T. gondii* antigens, we presumably enhanced the accessibility of relevant

epitopes, which led to significantly increased antibody levels and an improved assay
performance compared to a reference gold standard assay. Our advancements allow
measuring *T. gondii* antibodies reliably in large multi-pathogen panels and will help to conduct

[revised manuscript text omitted]

Skew-normal and Skew-T Mixture Models REVSTAT-Statistical Journal.

36. Domingues TD, Grabowska AD, Lee JS, Ameijeiras-Alonso J, Westermeier F,
Scheibenbogen C, Cliff JM, Nacul L, Lacerda EM, Mourino H, Sepulveda N. 2021.
Herpesviruses Serology Distinguishes Different Subgroups of Patients From the
United Kingdom Myalgic Encephalomyelitis/Chronic Fatigue Syndrome Biobank.
Front Med (Lausanne) 8:686736.

Tables

*Table 1: Characterization of T. gondii antigens*

Antigen	Amino acids	UniProt reference	Alternative names
SAG1	31* – 349	A0A125YP09	P30, SAG-related sequence SRS29B
SAG1D1	47 – 178		
SAG1C	301 – 333		
SAG1Pept	103 – 116		
P22	9* - 187	A0A125YIJ3	SAG2A, SAG-related sequence SRS34A
P22trunc	27 – 173		

494 *Signal peptides were removed

495 Table 2: Median antibody responses and IQRs against *T. gondii* antigens in individuals with positive (*T. gondii*
 496 pos.) and negative (*T. gondii* neg.) reference status determined in two serum dilutions.

		T. gondii pos.	T. gondii neg.	Average signal	AUC (95% CI)	p-value
		[MFI (IQR)]	[MFI (IQR)]	increase*		
1:100	SAG1	265 (165 – 407)	67 (44 – 119)	2.9	0.88 (0.82 – 0.94)	<0.001
	SAG1D1	991 (566 – 1,648)	49 (26 – 96)		0.98 (0.94 – 1.00)	
	P22	548 (238 – 1,240)	76 (41 – 123)	1.6	0.93 (0.89 – 0.97)	
	P22trunc	1,026 (454 – 2,133)	15 (7 – 27)		0.98 (0.97 – 1.00)	
1:1000	SAG1	141 (57 – 250)	18 (10 – 25)	4.4	0.93 (0.89 – 0.98)	<0.01
	SAG1D1	924 (296 – 1,813)	13 (5 – 21)		0.98 (0.95 – 1.00)	
	P22	187 (95 – 513)	16 (11 – 31)	2.3	0.94 (0.91 – 0.99)	
	P22trunc	481 (165 – 1,406)	4 (1 – 7)		0.99 (0.99 – 1.00)	

* Corresponds to regression line slopes

Table 3: Diagnostic accuracy applying threshold values based on (a) maximizing Youden's index after ROC analysis, (b) the local minimum of two mixed normal distributions and (c) the local minimum of two mixed skew-normal distributions

Serum dilution	Bead type	P22trunc				SAG1D1				
		Threshold [MFI]	Se (95% CI)	Sp (95%CI)	Cohen's κ (95%CI)	Threshold [MFI]	Se (95% CI)	Sp (95%CI)	Cohen's κ (95%CI)	
(a) ROC analyses										
1:100	A	70	94 (94 – 99)	95 (80 – 100)	0.86 (0.77 – 0.95)	141	98 (21 – 100)	93 (69 – 100)	0.92 (0.85 – 0.99)	
1:1,000	A	37	95 (89 – 100)	97 (87 – 100)	0.81 (0.71 – 0.91)	63	98 (38 – 100)	95 (84 – 100)	0.93 (0.86 – 1.00)	
1:1,000	B	158	95 (87 – 99)	98 (88 – 100)	0.89 (0.81 – 0.97)	394	99 (57 – 100)	93 (71 – 100)	0.93 (0.87 – 1.00)	
(b) Mixture of two normally distributed subpopulations										
1:100	A	113	93 (79 – 99)	95 (83 – 100)	0.83 (0.73 – 0.93)	180	98 (21 – 100)	93 (68 – 100)	0.90 (0.83 – 0.98)	
1:1,000	A	19*	96 (89 – 100)	95 (82 – 100)	0.81 (0.71 – 0.91)	220	82 (38 – 100)	97 (92 – 100)	0.67 (0.55 – 0.79)	
1:1,000	B	140	96 (90 – 100)	95 (83 – 100)	0.89 (0.81 – 0.97)	350	99 (58 – 100)	93 (71 – 100)	0.93 (0.87 – 1.00)	
(c) Mixture of two skew-normally distributed subpopulations										
1:100	A	95	93 (79 – 99)	95 (83 – 100)	0.83 (0.73 – 0.93)	165	98 (21 – 100)	93 (68 – 100)	0.90 (0.83 – 0.98)	
1:1,000	A	20*	96 (89 – 100)	95 (82 – 100)	0.81 (0.71 – 0.91)	55	98 (84 – 100)	92 (88 – 100)	0.93 (0.86 – 1.00)	
1:1,000	B	165	94 (93 – 99)	98 (90 – 100)	0.89 (0.81 – 0.97)	290	99 (84 – 100)	90 (71 – 100)	0.92 (0.84 – 0.99)	

* A minimal threshold of 30 MFI was applied corresponding to the lower limit of quantitation
 501 A: non-magnetic beads; B: magnetic beads
 Se: sensitivity [%]; Sp: specificity [%]; κ: Cohen's kappa; CI: confidence interval

Figures

Figure 1: Antibody levels against *T. gondii* antigens in seropositive (magenta) and seronegative individuals (blue), as well as their eligibility to distinguish them visualized as ROC curves in a serum dilution of 1:100 (A) and 1:1,000 (B).

Figure 2: Antibody responses against the *T. gondii* antigens SAG1D1 and P22trunc measured in multiplex serology assays using non-magnetic beads and magnetic beads. Dotted lines represent the lower limit of quantitation (30 MFI) using a 1:1,000 sample dilution. Magenta-colored dots represent individuals with a positive reference status for *T. gondii*, while blue dots represent individuals with a negative reference status for *T. gondii*.

Figure 3: Antibody levels against the T. gondii antigens SAG1D1 and P22trunc in individuals with a seropositive (magenta) and seronegative (blue) reference status. Lines represent proposed thresholds for seropositivity, derived from ROC analysis (solid), fitting a mixture of two normal distributions (dashed) or fitting a mixture of two skew-normal distributions (dotted).

Dear Reviewers,

Thank you for carefully reviewing our manuscript and providing us with constructive feedback and insightful comments. We appreciate each suggestion and believe that the quality of the manuscript has significantly improved. Please find a point-by-point reply to the issues raised below.

Sincerely, on behalf of all co-authors,

Dr. Rima Jeske

Review 1

Goal: Sero-epidemiological studies are a valuable approach to address open questions about risk factors, potential outcomes, trends, and co-infections in large cohorts of people.
Method: *T. gondii* multiplex serology approach, which is based on the immunodominant antigens SAG1 and P22.

Conclusions:

1. Using the truncated versions, SAG1D1 and P22trunc, significantly enhanced signal-to-noise ratios were achieved, and with almost perfect concordance with the gold-standard Sabin Feldman dye test. (Yes)
2. In sample dilutions of 1:100, the diagnostic accuracy of SAG1D1 and 29 P22trunc reached sensitivities of 98% and 94% and specificities of 93% and 95%, respectively. (Yes)
3. Importantly, however, these results were reproducible in a 1:1,000 sample dilution, using both magnetic and non-magnetic beads, and across multiple methods to determine thresholds for seropositivity, including receiver operating characteristic (ROC) analysis and finite mixture models. (Yes)
4. Our improved multiplex serology assay is therefore able to generate robust and reproducible results under various assay conditions, thus enabling inclusion of *T. gondii* antibody measurements with other pathogens, in large multiplex serology panels for sero epidemiological research. (No)
5. Our revised multiplex serology assay equips researchers with a powerful tool capable of delivering reliable and reproducible results under diverse assay conditions. This advancement paves the way for the integration of *T. gondii* antibody measurements into multi-pathogen multiplex serology panels, promising valuable insights into public health and pathogen interactions. (No)
6. Our advancements allow measuring *T. gondii* antibodies reliably in large multi-pathogen panels and will help to conduct sero-epidemiological research which is crucial for disease surveillance, risk assessment, and informing public health interventions. (No)

All sections highlighted red are not supported by the results. The authors state in the discussion section that the results “seem promising” but “require further validation”. Claims about reproducibility and reliability of results, as well as the ability to include this test in larger studies, should not yet be made until those tests or experiments are run and shown to be reliable and reproducible.

Thank you for carefully going through the manuscript and evaluating the core claims. We appreciate your concerns regarding upscaling, multiplexing and reproducibility and hope that we can address them accordingly.

Multiplex serology is the core technology of our research and we gained experience and routine over the years analyzing numerous large serological studies. One of the main advantages is that the technique allows combining antigens from different pathogens with each other while remaining a high reproducibility. Nevertheless, we make sure to validate every single antigen thoroughly to exclude cross-reactivity or adverse effects.

*In the preparation for a large seroepidemiological study which will comprise 40,000 human serum samples, the *T. gondii* reference samples were re-tested (a fourth time) in a 48-plex assay. This means that antibodies against the two *T. gondii* antigens were quantified alongside 46 antigens from other pathogens. Compared to the monoplex assay (2 *T. gondii* antigens + 1 control antigen), Pearson's *r* reached 0.98 for SAG1D1 and 0.97 for P22trunc (Figure below).*

*Hence, the integration of the two *T. gondii* antigens into our multiplexed platform works very well. These results also support the claims that assay results are highly reproducible, which we also found for other antigens as shown in the assay validation for the UK biobank (reference 12).*

Nevertheless, we agree that a modification of the manuscript text is necessary, as these results are not included in the present validation.

4. was changed to:

*Our improved multiplex serology assay is therefore able to generate robust and reproducible performance metrics under various assay conditions. Inclusion of *T. gondii* antibody measurements with other pathogens, in multiplex serology panels will allow for large-scale sero-epidemiological research.*

5. was changed to:

*Our revised multiplex serology assay equips researchers with a powerful tool capable of delivering *T. gondii* serum antibody measurements with high sensitivity and specificity under diverse assay conditions. This advancement paves the way for the integration of *T. gondii* antibody measurements into multi-pathogen multiplex serology panels, promising valuable insights into public health and pathogen interactions.*

6. was changed to:

*Our advancements will allow integrating *T. gondii* antibody measurements into large multi-pathogen panels, as demonstrated for other pathogens in the UKB and the China Kadoorie Biobank (12, 39). This will facilitate sero-epidemiological research which is crucial for disease surveillance, risk assessment, and informing public health interventions.*

Comments:

Line 14: Yet, our understanding of risk factors for infection and disease, potential outcomes and their trends, and the potential role of co-infections is still limited.

- Understanding is not limited. Much is understood from previous studies. Focus more on how increased understanding is still valuable and how it could be used.

While there has been substantial progress in the molecular and infection biology field, epidemiological studies (especially on humans) are very scarce, e.g. potential associations with cancer or neurobehavioral changes are still ambiguous and usually only contain few individuals.

*To put the focus on the epidemiological side, the sentence was adapted as follows:
'Yet, our understanding of long-term consequences, associated risk factors and the potential role of co-infections is still limited. Sero-epidemiological studies are a valuable approach to address open questions and enhance our insights into T. gondii in human populations.'*

Line 29: P22trunc reached sensitivities of 98% and 94% and specificities of 93% and 95%, respectively.

- "Sensitivities" and "specificities" have not previously been defined and it would be helpful to clearly state what they signify or mean.

The text was adapted to:

'In sample dilutions of 1:100, the diagnostic accuracy of SAG1D1 and P22trunc reached sensitivities (true positive rates) of 98% and 94% and specificities (true negative rates) of 93% and 95%, respectively.'

Line 30: Importantly, however, these results were reproducible in a 1:1,000 sample dilution, using both

-Delete "however"

-Specify which results were reproducible.

The sentence was adapted as follows:

'Importantly, performance metrics were reproducible in a 1:1,000 sample dilution, using both magnetic and non-magnetic beads, and across multiple methods to determine thresholds for seropositivity, including receiver operating characteristic (ROC) analysis and finite mixture models.'

Line 55: The major source of these environmental contaminations is suggested to be domestic cats (4, 5).

- Mentioning that infected cats are the definitive host for the parasite and cause this contamination through shedding of oocysts in feces gives the reader a better understanding of how the parasite spreads.

We added the information. The sentence is now as follows:

'The major source of these environmental contaminations is suggested to be domestic cats due to the excretion of oocysts in their feces'

Line 58: In congenitally infected newborns and immunocompromised individuals, e.g. tissue transplanted or HIV-infected individuals, T. gondii may proliferate, resulting in a range of severe or life-threatening conditions, e.g. encephalitis (1).

- Consider including the understanding that both acute and latent chronic infections can

escape immune control and be life-threatening to immunocompromised individuals. This would help the transition to, and giving more weight to, the following paragraph.

*Thank you for pointing this out. We added your suggestion as follows:
'In congenitally infected newborns and immunocompromised individuals, e.g. tissue-transplanted or HIV-infected individuals, acute and chronic T. gondii may escape immune control, resulting in a range of severe or life-threatening conditions, e.g. encephalitis.'*

Line 102: All samples had been stored at -20°C.

- Wonderful. This adds value to the study showing the robustness of the method, allowing multiple freeze/thaw cycles of sample according to previous publications.

We try to avoid freeze/thaw cycles whenever possible and especially when dealing with large study cohorts. In some cases, e.g. when samples are retested, this is obviously not possible. It is indeed reassuring to know that the assay is still robust.

Line 96: Serum samples for the validation of the T. gondii multiplex serology assay were kindly provided by the Toxoplasma Reference Unit of Public Health Wales (Prof. Edward Guy)

- More information on these samples would be important. For example, where were the samples collected and is it known what strains or "(types I, II, III)" the infected individuals were infected with? I see that some of this information is included in other papers, but referencing a few of the features of this data set is relevant.

- This would also lead into the needed important background on the expression of each of these antigens being tested in the different T. gondii strains, as prevalence of the different strains varies greatly among locations and populations being tested. I would strongly suggest citing some of John Boothroyd's lab's recent work here for the introduction.

All sera derived from UK residents and were submitted to the Toxoplasma Reference Unit for routine toxoplasma serological testing. They comprise a proportion of individuals who will have acquired toxoplasma infection while living in the UK, together with a number of other EU citizens living in the UK and immigrants from outside the EU with a lifelong immune response against strains of toxoplasma acquired in their country of origin. While T. gondii strain are not routinely genotyped, most of them are presumable of the type II, as this is the most common one in Europe/UK.

Serological assays are not capable to distinguish between strains of different types, as the cross-reactivity between antigens is too high. Prof. Boothroyd's lab published an approach based on polymorphic peptides to overcome this, however, it lacks validation in humans.

Nevertheless, our aim is enabling a high-throughput testing for T. gondii infection in large patient cohorts, which universally works for all T. gondii types. Hence, we used antigens that are expressed in virtually all strains and also chose highly conserved regions.

We adapted the manuscript as follows:

Methods:

'Serum samples for the validation of the T. gondii multiplex serology assay were obtained through routine testing and kindly provided by the Toxoplasma Reference Unit of Public Health Wales (Prof. Edward Guy).

Discussion:

Our multiplex serology assay is based on the two immunodominant proteins P22 (SAG2A) and SAG1 (P30), which are expressed in virtually all T. gondii strains. [Theisen, Terence C.,

and John C. Boothroyd. "Transcriptional signatures of clonally derived Toxoplasma tachyzoites reveal novel insights into the expression of a family of surface proteins." *Plos one* 17.2 (2022): e0262374.]

Line 187: Overall, the signal-to-noise ratio was increased for both truncated antigens with MFI values reaching on average 2.9-times higher MFI values for SAG1D1 and 1.6-times for P22trunc compared to the respective full-length versions (Table 2, Figure 1).

- This statement is accurate, but when working with MFI referencing fold changes instead of raw numerical MFI changes can be misleading. For example, the difference in an MFI of 10 and 30 results in a 3-fold difference but only a 20 unit difference. The difference in an MFI of 50 and 100 results in a 2-fold difference but 50-unit difference. This exact scenario occurs in the next comment. The larger raw unit difference can be what matters most for being able to differentiate between real signal and noise in most experiments. If this is not the case for the models being run, then an explanation of this should be included.

We describe raw numbers in the paragraph before and include them in Table 2 and visualize them Figure 1. Nevertheless, you raise a valid concern regarding the potential for misleading interpretation when reporting fold change values for net MFI numbers.

In order to address the raw unit difference between signal and noise, we decided to report the signal-to-noise ratio increase instead of MFI value increase and adapted the text and table accordingly.

It now reads as follows:

'Compared to the respective full-length antigen, the signal-to-noise ratio was increased 5.1-fold for SAG1D and 9.5-fold for P22trunc (Table 2, Figure 1).'

We also made sure to add this information in the method section:

Signal-to-noise ratios were obtained by dividing the median net MFI of individuals with a positive reference status (signal) by the median net MFI of those with a negative reference status (noise).

Line 199: Linear regression revealed that the difference in antibody levels between full-length and truncated antigens was even more prominent than in the 1:100 serum dilution, with MFI values being on average 4.4-times increased for SAG1D1 and 2.3-times for P22trunc compared to the respective full-length versions (Table 2, Figure 1). Correlations were also higher and reached Pearson's r values of 0.86 and 0.95, respectively (Table 2).

- It is okay to point out that the fold difference is greater in the 1:1000 dilution, but the raw difference in MFI is only greater for SAG1D1 and not P22trunc. Neglecting this point can be misleading.

As described above, we decided to switch to reporting signal-to-noise ratio increase.

The paragraph was changed as follows:

Hence, the signal-to-noise ratio was increased 9.1-fold for SAG1D1 and 10.3-fold for P22trunc and even more prominent than in the 1:100 serum dilution (Table 2, Figure 1). Correlations were also high and reached Pearson's r values of 0.86 and 0.95, respectively (Table 2).

Line 217: Using non-magnetic beads, median antibody responses reached 924 MFI (IQR: 296 – 1,813 MFI) against SAG1D1 and 481 MFI (IQR: 164 – 1,479 MFI) against P22trunc in seropositive individuals (Table 2). On magnetic beads, antibody levels were increased 6.2-times with a median reaching 5,534 MFI (IQR: 1,850 – 10,183 MFI) and 4.0-times with a

median of 2,230 MFI (IQR: 730 – 5,714 MFI), respectively, as extracted from the regression line slopes. This led to all antibody responses in seropositive individuals to exceed the lower limit of quantitation of 30 MFI (Figure 2).

- The same statement stands for fold vs. raw unit increases.

In this paragraph, we want to emphasize that MFI values are systematically increased (for sero-positive and sero-negative individuals) when using magnetic beads. Hence, we report regression line slopes and also median MFI values which are reached for both sero-positive and sero-negative individuals in order to emphasize this. The raw unit increase is only of secondary importance here.

Discussion

Line 267: These results were reproducible using different bead types and methods to determine thresholds for seropositivity with Cohen's kappa being consistently above 0.9.

- It is clear that ROC analysis was done and finite mixture models were created, but previous papers do go into more detail about these methods than this paper does, and so some questions still remain. Was the model to determine threshold for seropositivity developed using only a subset of the samples and then tested on the remaining samples, or does the model determining the threshold for seropositivity still need to be tested using new samples to measure how accurately it can determine positive from negative samples? Further statements in the discussion suggest that the model has not been tested on other samples, but if that is incorrect then further explanation is needed.

With the aim to validate a specific threshold (x MFI), it is a common practice to separate samples into a training set to define an optimal threshold, and a validation data set to estimate how well this threshold performs. In a next step, a secondary validation with external samples can be performed to assess the generalizability of the specified threshold.

However, numerous studies and publications have shown that pre-specified thresholds do not perform equally well across different populations (e.g. Kafatos G et al: Is it appropriate to use fixed assay cut-offs for estimating seroprevalence?). Reasons include variances in background immunity in certain populations, differing sample preparations or simply different Lot numbers across assay chemicals.

Hence, a method which generates thresholds just based on the distribution of the results, without considering external reference samples, could be superior. We test this in our samples by applying the same algorithm while modifying assay parameters (e.g. sample dilution, bead type) and consequently generating different MFI values.

The EM algorithm solely considers the data distribution and fits a bimodal density curve, which has two subpopulations and a local minimum in between. If these subpopulations actually correspond to sero-positive and sero-negative individuals is determined in a next step comparing the results to the gold-standard and determining sensitivity and specificity.

Here, we show that we can achieve assay parameters which perform equally well to ROC-derived thresholds in a given population. These thresholds will differ in other studies due to study-specific setting which cannot be estimated beforehand (especially using cohort samples, as these often have only very limited volume).

Further validation studies are needed to find out if this bimodal distribution can also be seen in other populations. Due to the nature of the immunological response, this is highly likely as

we only expect sero-positives and sero-negatives. Based on the improved signal-to-noise ratio of the new antigens these two populations will now be more easily distinguishable, even in a 1:1000 dilution.

We made adaptations to the manuscript text to address this issue:

Methods:

'In order to compare multiplex serology results to the gold-standard Sabin-Feldman dye test, continuous MFI values were dichotomized using antigen-specific thresholds. These thresholds derived from finite mixture models, using an expectation-maximization (EM) algorithm to fit a bimodal distribution after log-transforming MFI values (25, 26). Thresholds corresponded to the local minima of the density curves. Sensitivities and specificities were determined in a next step based on the reference status of the sera.

To evaluate algorithm performance, the thresholds were compared with those derived from the classical maximization of Youden's index (receiver operating characteristic (ROC) analysis).'

Discussion:

'Our results showed that thresholds derived from finite mixture models seem promising to generate thresholds for T. gondii multiplex serology, which are solely based on the data distribution of the results. Nevertheless, a bimodal data distribution needs to be verified in further upcoming seroepidemiological the exploratory approach requires further validation in large-scale studies.'

Line 259: As *T. gondii* usually persists in a latent form, antibody levels decline but do usually not disappear.

- If desired, the authors can support this statement with background on how infection will continually go through small spontaneously reactivations from cysts, making sero-reversion more than exceedingly rare.

Thank you for this excellent suggestion. We included the information and cited Rougier et al: Lifelong Persistence of Toxoplasma Cysts: A Questionable Dogma?

Line 362: Although, our results showed that thresholds derived from finite mixture models seem promising for *T. gondii* multiplex serology, the exploratory approach requires further validation in large-scale studies. Modeling thresholds might also be transferrable to other antigens if both subpopulations (seropositive and seronegative individuals) are large enough. Similarly, validation studies for all antigens are essential, as antibody kinetics might differ and be more complex due to reinfection rates, cross-reactivity, population differences, age-dependence, and others (18, 35, 36).

- This statement is accurate, but it contradicts the previously and subsequent conclusion that the method generates "robust and reliable results" that could help conduct sero-epidemiological research. Either training the model on a subset of the already collected data and testing it on the other subset, or ideally testing the current threshold models on new samples, is needed to claim that the method can reliably and robustly generate data usable to conduct sero-epidemiological research. It is also acceptable to remove these claims, or change them and instead focus on the proven improvements on assay sensitivity. This could still include a proposed threshold for sero-positivity that could be tested in future experiments. That is already a significant finding.

- Either delete "and others", or include "variables such as" before "reinfection rates".

This question is addressed in the previous answer. We changed the specific paragraph slightly, to put the emphasize more on the method than on the thresholds:

*Our results showed that finite mixture models seem promising to generate thresholds for *T. gondii* multiplex serology, which are solely based on the data distribution of the results. Nevertheless, a bimodal data distribution needs to be verified in upcoming seroepidemiological studies. Modeling thresholds might also be transferrable to other antigens and pathogens if both subpopulations (seropositive and seronegative individuals) are large enough. Similarly, validation studies for all antigens are essential, as antibody kinetics might differ and be more complex than for *T. gondii* due to variables such as reinfection rates, cross-reactivity, population differences, age-dependence, and others (18, 26, 38).*

Review 2

Dear Author Very good paper but need to some changes,

* Add Key words after Abstract.

The key-words 'Multiplex Serology', 'Toxoplasma gondii', 'Seroepidemiology' and 'High-throughput serology' were added

* In Title Gondii antigens page 5 line 123 from (These envelope glycoproteins to line 125 delete

Control antigens were added to the measurements to ensure comparability between runs. We are convinced that these antigens are crucial to claim that the difference between full-length and truncated antigens is not just sheer coincidence.

* Add conclusion after Discussion and in Abstract.

Conclusion parts seems not to be common in Microbiology spectrum.

* Add Author affiliation.

Author affiliation were added when uploading the manuscript to the journal's website.

* Add Author Orchid.

Orchid IDs will be added to the final version.

* Add Funding.

The funding information was added to the submission process.

* Add Author Contributions.

We added the author contributions to the re-submission.

Re: Spectrum03618-23R1 (Advancing *Toxoplasma gondii* multiplex serology)

Dear Dr. Rima Jeske:

Your manuscript has been accepted, and I am forwarding it to the ASM production staff for publication. Your paper will first be checked to make sure all elements meet the technical requirements. ASM staff will contact you if anything needs to be revised before copyediting and production can begin. Otherwise, you will be notified when your proofs are ready to be viewed.

Sincerely,
Mark Pandori
Editor
Microbiology Spectrum